# Application Research on High-Precision Tiltmeter with Rapid Deployment Capability

**DOI:** 10.3390/s25051559

**Published:** 2025-03-03

**Authors:** Fuxi Yang, Dongxiao Guan, Xiaodong Li, Chen Dou

**Affiliations:** 1Xinjiang Seismological Bureau, Urumqi 830011, China; guandongxiao3720@163.com (D.G.); lxdxj12345@163.com (X.L.); 2School of Emergency Management Science and Engineering, University of Chinese Academy of Sciences, Beijing 100049, China; douchen22@mails.ucas.ac.cn

**Keywords:** vertical pendulum tiltmeter, high precision, crossing plate

## Abstract

This article introduces a high-precision vertical pendulum tiltmeter with rapid deployment capability to improve the observation efficiency, practicality, and reliability of geophysical site tilt observation instruments. The system consists of a pendulum body, a triangular platform, a locking pendulum motor, a sealed cover, a ratio measurement bridge, a high-precision ADC, and an embedded data acquisition unit. The sensing unit adopts a vertical pendulum system suspended by a cross spring and a differential capacitance bridge measurement circuit, which can simultaneously measure two orthogonal directions of ground tilt. The pendulum is installed on a short baseline triangular platform, sealed as a whole with the platform, and equipped with a locking pendulum motor. When the pendulum is locked and packaged, it can withstand a 2 m free fall impact, with high reliability and easy use. It can be quickly deployed without the need for professional technicians. This article analyzes its various performance and technical indicators based on its application in the rapid deployment of the Zeketai seismic station in Xinjiang. It is of great significance for emergency response, mobile observation, base detection, anomaly verification, and other applications of ground tilt.

## 1. Introduction

Crustal deformation represents a critical area of study within geophysics, focusing on the observation of crustal movements and the characterization of associated deformation processes and their evolution over time. Data from crustal deformation observations have extensive applications in geodesy and geophysics, providing valuable insights into earthquake processes and potential precursors. These observations form a scientific basis for earthquake monitoring and forecasting efforts. Since the 1980s, several countries including the United States, China, Japan, and Russia have launched modern research programs to study crustal movements, with a particular emphasis on plate tectonics, tectonic deformation, and the effects of solid earth tides. Crustal deformation observations are typically categorized into four types: spatial deformation measurements, regional deformation measurements, fault deformation measurements, and solid tide deformation measurements [1,2,3]. The monitoring of solid tidal deformation focuses on the Earth’s response to lunar and solar tidal forces and comprises four primary components: ground tilt, gravity, cavity strain, and borehole strain. Ground tilt observations, which analyze vertical relative motion and dynamic changes in solid tides, employ techniques based on both vertical and horizontal references. Given the subtle diurnal variations associated with solid tides, tiltmeter sensitivity must achieve a resolution of 0.0002″.

In long-term observations of ground tilt, the technology and performance of inclinometers have continuously evolved to meet the demands for high-precision measurements. Currently, the deformation monitoring network primarily uses types such as quartz horizontal pendulum inclinometers, DSQ-type water tube inclinometers, and VS-type or VP-type vertical pendulum inclinometers [4,5,6,7].

The horizontal pendulum inclinometer originated from a double-wire suspension system developed in 1830, which, after improvements, could be used for measuring seismic and slow ground movements. In the early 1970s, China developed the SQ-70D photographic recording quartz horizontal pendulum inclinometer, followed by the SSQ-1 digital magnetic tape-recording horizontal pendulum inclinometer in the 1990s [8], and the SSQ-2I horizontal pendulum inclinometer in the early 21st century. Supported by the Chinese earthquake monitoring network construction project implemented during the 10th Five-Year Plan, more than sixty quartz horizontal pendulum inclinometers have been installed in China’s current deformation monitoring network [9]. The measurement accuracy of quartz horizontal pendulums is related to the swing period, and their structure typically employs a double-suspension wire system, which provides a high magnification factor. However, this design results in a large volume for the horizontal pendulum. Additionally, the measurement period can be affected by the self-vibration period of the suspension wires, necessitating frequent calibration, which is challenging due to the limitations of the measurement mechanism.

The water tube inclinometer was first proposed by Michelson and Gale in 1914, using optical interference methods to measure displacements at both ends of the water tube, later applied in geodetic leveling. Since its development in 1998, the DSQ-type water tube inclinometer has been installed and used in over one hundred networks in China for more than twenty years, as part of a subproject under the key project “Development of Medium and Short-term Precursors Observation Instruments” by the China Earthquake Administration [10]. The measurement accuracy of the DSQ-type water tube inclinometer increases with longer baseline lengths, typically ranging from 20 to 30 m, resulting in a large device primarily suitable for measuring average tilt over large areas, making it difficult to conduct fine observations at specific points and introducing multiple sources of error. Furthermore, over time, the water in the tube may deteriorate, necessitating frequent water changes to ensure measurement accuracy, complicating maintenance.

The vertical pendulum inclinometer measures the offset of the crust relative to the vertical line, using the plumb line as a reference. Early vertical pendulum inclinometers had longer pendulum lengths due to the small changes in ground tilt. With the development of high-precision capacitive displacement sensors, pendulum lengths have significantly shortened, leading to increased miniaturization, making them predominant in ground tilt measurements. In 1997, the VS-type vertical pendulum inclinometer was developed as a key instrument project by the China Earthquake Administration, passing expert acceptance in June 1999 [11]. In 2010, Professor Hu Guoqing led a team to develop the VP-type vertical pendulum inclinometer. This instrument uses capacitive displacement sensors as the measuring unit, improving the measurement bandwidth compared to the VS-type [12]. However, the VP single-component vertical pendulum inclinometer is relatively large and requires two different instruments placed orthogonally to measure components in different directions.

All of the above high-precision inclinometers face challenges such as fragile sensor units and complex installation processes, requiring professional and experienced engineering technicians for on-site assembly, debugging, and necessary trial runs before formal observations can commence. Thus, high-precision inclinometers are limited in fields that require rapid deployment and mobile measurements, such as emergency responses to major earthquakes, dense observations, and verification of seismic anomalies [13].

This paper introduces a high-precision component vertical pendulum inclinometer—ICT-1 (Integration Capacitive Tiltmeter)—which employs a cross spring as the suspension structure. This design not only enhances the strength of the pendulum body but also allows a single pendulum system to simultaneously measure multiple components of tilt, improving measurement efficiency and reducing the device’s volume. To address the practical needs of mobile observations and emergency measurements, the ICT-1 inclinometer integrates several innovative features in its design: the introduction of a locking mechanism provides reliable protection for the pendulum body; the design of a short baseline triangular platform significantly optimizes the device’s dimensions; the built-in zeroing motor and overall sealed structure enhance the instrument’s shock resistance, making it suitable for ordinary transportation conditions; simultaneously, the integrated design avoids the complexity of on-site assembly, achieving rapid deployment. Through the integrated design of the tilting pendulum system, the modified ICT integrated chamber inclinometer presented in this paper is highly reliable, adaptable, and user-friendly, allowing for quick deployment without the need for specialized technicians. It can accurately record solid tidal responses and co-seismic responses, which are of great significance for observing geophysical site deformation.

## 2. ICT-1 Integrated Chamber Inclinometer

The ICT-1 integrated chamber inclinometer consists of an integrated pendulum, a front box, and a data collector. As shown in Figure 1, the vertical pendulum body is a tilt sensing unit installed on a triangular platform. The platform can be balanced and adjusted by a high-precision deceleration stepper motor. The pendulum body is equipped with a locking motor, which can be pushed and locked during transportation or non-working conditions. Tilt measurement uses differential capacitance sensing circuits, and the measurement and motor drive circuits are installed in the front box, which can be manually or remotely controlled [14,15,16]. The front box converts the tilt amount of two components into voltage signals and outputs them to the data collector. The data collector performs AD conversion and data storage, and can transmit data through FTP, HTTP, or dedicated network protocols [17]. The data collector simultaneously listens to network commands and is responsible for receiving calibration, balance adjustment, and other commands sent by network users, achieving remote control of the pendulum by users. Figure 2 shows the installation site comparison of the ICT-1 integrated tiltmeter, VP single-component vertical pendulum tiltmeter, quartz horizontal pendulum tiltmeter, and DSQ-type water pipe tiltmeter used in this study. It can be seen that, on the base platform of 100 cm × 80 cm, the dipmeter in this study has a small volume, and only one instrument is needed to measure the north–south and east–west components, while the VP single-component vertical pendulum dipmeter has a large volume, and two different instruments are required to be placed orthogonally to measure the components in different directions. On the base platform of 150 cm × 100 cm, the volume of the quartz horizontal pendulum tiltmeter is significantly larger than that of the ICT-1 vertical pendulum tiltmeter, and the DSQ water pipe tiltmeter occupies a larger area, with a base length of 20 m.

### 2.1. Integrated Pendulum and Triangular Platform

The vertical pendulum tiltmeter typically employs a dual-leaf spring suspension structure, as shown in Figure 3a. In this configuration, the pendulum body is suspended by leaf springs on both sides, forming a vertical unidirectional pendulum. This suspension mechanism effectively constrains the pendulum body, allowing simple harmonic oscillation exclusively in the measurement direction while preventing torsion. When horizontally positioned, oscillations in the non-measurement direction are minimal, resulting in high sensitivity for unidirectional tilt measurements. This design is commonly utilized in high-precision pendulum tiltmeters. However, it is unsuitable for rapid deployment or portable emergency observation equipment.

While the unidirectional pendulum design facilitates flexible oscillation in the measurement direction, large tilt angles or significant impacts in the non-measurement direction can cause uneven forces on the leaf springs, leading to bending, fracture, or plastic deformation. To ensure proper operation, the pendulum device must be precisely leveled in all directions.

The ideal pendulum body for a vertical pendulum inclinometer is suspended by a single thread, allowing it to swing in any direction. However, when a pendulum body suspended by a single thread is subjected to external forces or vibrations, changes in gravitational components can cause its swinging trajectory to deviate from the ideal simple harmonic motion model, resulting in twisting and rotating phenomena. To overcome the rotational twisting of the pendulum body, a dual-thread suspension can be used. With a dual-thread suspension, the pendulum body behaves like a unidirectional pendulum, swinging freely only in the direction of freedom. However, when the pendulum body tilts in a non-free direction, uneven force on the threads causes it to revert to a single-thread pendulum, leading to tilting and twisting of the pendulum body itself. Therefore, in practical applications, spring strips are used instead of threads, allowing the pendulum body to swing freely in the direction of freedom, while maintaining stability in the non-free direction due to the support of the spring strips, preventing oscillation in that direction. This is the commonly used unidirectional vertical pendulum body. Currently, the VS-type and VP-type vertical pendulum inclinometers used in geophysical networks adopt this structure. To ensure the sensitivity of the tilt measurement, the spring strips for the unidirectional pendulum must be sufficiently flexible, typically made of high-elasticity beryllium bronze, with a thickness not exceeding 0.1 mm. Such spring strips are very fragile; when the pendulum body tilts in a non-free direction, the supporting effect of the spring strips is limited, and the stress state of this support is very complex, akin to using an upright piece of paper to support a heavy object. Consequently, when the pendulum body tilts excessively in a non-free direction or experiences minor vertical bouncing during disturbances, even with a locking mechanism installed, it is easy for the spring strips to become damaged. Sometimes, the locking operation itself can also harm the strips, as shown in Figure 3b. This fragility and susceptibility to damage result in poor impact resistance, necessitating extreme care during transportation, installation, and use. In many cases, to ensure the safety of the equipment, technicians may even install the pendulum body and spring strips at the observation site, significantly limiting their application in emergency monitoring and mobile measurement fields that require rapid deployment and transitions. This is also why high-precision chamber tilt observations are classified as point deformation observations. In people’s minds, high-precision vertical pendulum inclinometers and water tube inclinometers, as well as quartz horizontal pendulum inclinometers, can only be used for fixed-point observations after installation.

To address these limitations, the ICT-1 integrated chamber tiltmeter incorporates a cross-spring pendulum system, as shown in Figure 3c. This system features two single-degree-of-freedom dual-suspension leaf springs arranged in orthogonal superposition. The cross-spring design maintains the sensitivity and anti-torsion capabilities of the dual-leaf spring suspension while enabling free oscillation in both orthogonal directions. This allows simultaneous measurement of tilt in the north–south and east–west directions. The sensitivity of the cross spring and the unidirectional spring blade for tilt measurement is comparable. Its advantage is primarily the reduction in equipment size, allowing a single pendulum body to measure tilt in two or more directions. Furthermore, the pendulum body can undergo significant oscillations in response to large tilts or impacts without damaging the springs.

After the spring blade is damaged, slight damage can cause a skipping phenomenon, while severe damage may lead to the device not functioning properly. Figure 4 shows the skipping phenomenon that occurred after the installation of the CBT-1 inclinometer using a unidirectional spring blade at the Xinyuan seismic station in Xinjiang. After maintenance and testing, it was confirmed that during installation, the device was tilted excessively in a non-measurement direction and experienced a certain impact, causing the spring blade to undergo inelastic deformation, resulting in a crease, as shown in Figure 5.

The orthogonal superposition of oscillation enables unrestricted motion, enhancing the system’s robustness. Compared to vertical unidirectional pendulum tiltmeters, this design reduces equipment size, improves impact resistance, and is better suited for portable installation and rapid deployment.

The cross-spring system can be viewed as the orthogonal superposition of two unidirectional pendulums. In a single direction, its motion can be analyzed as a unidirectional pendulum, as shown in Figure 3d. Assuming the pendulum body is a point mass, and the springs and pendulum rods are weightless, the dynamic equation describing the pendulum’s motion when deviating from equilibrium is given by:(1)$$mL\frac{d^{2}\theta }{dt^{2}}=-mgsin\theta \;\mathrm{or}\;\frac{d^{2}\theta }{dt^{2}}=-\frac{g}{L}sin\theta$$
where *θ* is the pendulum angle, *L* is the pendulum length, and *m* is the mass of the pendulum body. For an initial phase angle $\theta_{0}$ and defining $\omega^{2}=\frac{g}{L}$, the exact oscillation period is:(2)$$T=2\pi \sqrt{\frac{L}{g}}\cdot \left[1+\frac{1}{2}{sin}^{2}\frac{\theta_{0}}{2}+\cdots +\frac{1\cdot 3^{2}\cdot 5^{2}\cdots {\left(2n-1\right)}^{2}}{2^{2}\cdot 4^{2}\cdot 6^{2}\cdots {\left(2n\right)}^{2}}{sin}^{2n}\frac{\theta_{0}}{2}+\cdots \right]$$

This shows that the oscillation period increases with the initial phase angle. When $\theta_{0}\to 0$, higher-order terms (${sin}^{2n}\frac{\theta_{0}}{2}$) can be neglected, and the motion approximates simple harmonic oscillation.

When considering additional factors, such as the mass of the pendulum body and rod, and leaf springs, air damping, and buoyancy, and assuming a regular shape with uniform mass distribution, the dynamic equation becomes:(3)$${(J}_{B}+mL^{2}+J_{S})\frac{d^{2}\theta }{dt^{2}}=-mgsin\theta -\frac{m_{0}g}{2}\left(L-d\right)sin\theta +\rho_{0}VgLsin\theta$$
where $J_{B}$ is the moment of inertia of the pendulum body around its center of gravity; $J_{S}$ is the moment of inertia of the spring around its suspension point; *L* is the pendulum length; *d* is the distance from the upper suspension point to the center of gravity of the pendulum body; *m* and $m_{0}$ are the masses of the pendulum body and rod, respectively; $\rho_{0}\;$ is the air density; and *V* is the volume of the pendulum body, where V=m/ρ (as shown in Figure 3e). For small angles, when (*θ*→0), the equation simplifies to:(4)$$\frac{d^{2}\theta }{dt^{2}}+\frac{mgL+\frac{m_{0}g}{2}\left(L-d\right)-\frac{\rho_{0}}{\rho }mgL}{J_{B}+mL^{2}+J_{S}}\cdot \theta =0$$

The oscillation period becomes:(5)$$T=2\pi \sqrt{\frac{L}{g}\cdot \frac{1+\frac{J_{B}}{mL^{2}}+\frac{J_{S}}{mL^{2}}}{1+\frac{m_{0}}{2m}\left(1-\frac{d}{L}\right)-\frac{\rho_{0}}{\rho }}}=2\pi \sqrt{\frac{L}{g}\cdot \frac{A}{B}}$$
where *A* and *B* are constants. Considering the influence of these factors, the period becomes:(6)$$T=2\pi \sqrt{\frac{L}{g}\cdot \frac{A}{B}}(1+\frac{1}{2}{sin}^{2}\frac{\theta_{0}}{2}+\cdots )$$

The oscillation equation at this time is:(7)$$\theta =\theta_{0}\cos\left(\frac{2\pi }{T}\cdot t+\alpha \right)\;\mathrm{or}\;\theta =\theta_{0}\cos\left(\omega_{0}t+\alpha \right)$$

Considering air resistance, the motion becomes damped. At low velocities, the resistance is proportional to velocity, resulting in underdamped oscillations:(8)$$\theta =\theta_{0}\cdot e^{-\frac{\sqrt{1-\epsilon^{2}}}{\epsilon }\cdot t}\cdot \cos\left(\omega \cdot t+\alpha \right)$$
where $\omega =\omega_{0}\sqrt{1-\epsilon^{2}}$ is the underdamped oscillation frequency, and *ϵ* is the damping ratio. Incorporating damping effects, the corrected oscillation period becomes:(9)$$T=2\pi \sqrt{\frac{1}{1-\epsilon^{2}}\cdot \frac{L}{g}\cdot \frac{A}{B}}(1+\frac{1}{2}{sin}^{2}\frac{\theta_{0}}{2}+\cdots )$$

### 2.2. Tiltmeter Measurement and Control Circuit

High-precision tilt measurement can be achieved using various sensing circuits such as differential capacitance, CCD optical lever, eddy current sensors, and fiber Bragg gratings. Quartz horizontal pendulum tiltmeters, which are widely used in geophysical networks, typically employ CCD optical levers or eddy current sensors. Some vertical pendulum tiltmeters use fiber Bragg grating sensors, while others, such as VS and VP vertical pendulum tiltmeters, use differential capacitance sensors. These sensing circuits can all meet the requirements for high-precision tilt measurement, but each has its own advantages and disadvantages. The ICT-1 integrated chamber tiltmeter adopts a differential capacitance-sensing circuit, which involves a comprehensive consideration of cost, size, and portability.

The CCD optical lever sensor requires optical path amplification, so field installation involves placing a reflector and focusing the optical path, which occupies significant space. This reduces the device’s adaptability to different sites and hinders rapid deployment. Fiber Bragg grating sensors have high demodulator costs and are more suitable for distributed multi-sensor measurements, making them less cost-effective for single-point, single-device applications. Eddy current sensors are low-cost, compact, and easy to integrate, but their sensitivity is relatively poor. They require mechanical amplification to meet measurement requirements, making them suitable for horizontal pendulum tiltmeters with over a thousand times mechanical amplification. However, vertical pendulum tiltmeters have a simpler structure, and increasing the pendulum length for mechanical amplification is not conducive to device miniaturization.

In summary, the differential capacitance-sensing circuit offers high measurement accuracy, compact size, ease of integration, and relatively simple circuitry at a lower cost, making it well-suited for vertical pendulum chamber tiltmeters.

The ICT-1 integrated chamber inclinometer adopts a cross-spring pendulum body, with two sets of capacitor plates on each side of the pendulum, forming two sets of differential capacitors with the pendulum. The pendulum serves as the middle plate shared by the two sets of differential capacitors. Due to the shared use of a pendulum as the center plate by the two bridge circuits, different frequency excitation signals are used to distinguish between them [18]. The two excitation signals have a frequency doubling relationship, with the north–south component excitation using a frequency of 1.56 kHz and the east–west component excitation using a frequency of 3.12 kHz. Two sets of differential capacitors are connected to a transformer grounded at the center tap to form a measuring bridge. The unbalanced signal caused by the tilt change of the bridge is superimposed on the pendulum, amplified, and extracted by a phase-sensitive detection circuit at their respective frequencies. The extraction of phase-sensitive detection signals is shown in Figure 6. Two sets of phase-sensitive detection circuits use reference signals from the same source as the excitation for detection, and after low-pass filtering, extract their respective DC voltage signals.

### 2.3. Rapid Deployment Capability of Integrated Chamber Inclinometer

The integrated design of the ICT-1 type chamber inclinometer enables it to have high rapid deployment capability. When deploying it, first place the pendulum on the observation pier and align the direction indicator arrow of the pendulum with true north. At this time, the tilt sensors of the two components of the pendulum are used, with component 1 measuring the north–south tilt and component 2 measuring the east–west tilt. Connect the pendulum body, front box, and data collector with signal lines. After the data collector is powered on, it will supply power to the front box through the signal line. The front box is equipped with a lock pendulum switch. When the lock pendulum switch is turned on, the pendulum body enters the working state. The front panel LCD display shows the voltage values of two components for tilt measurement. According to the output voltage indication, manually adjust the foot screws of the triangular platform. When the voltage value is too high in the positive direction, rotate the corresponding foot screws counterclockwise. When the voltage value is too high in the negative direction, rotate the corresponding foot screws clockwise until the output voltage values of both components are close to 0 V. At this point, the pendulum is close to equilibrium and can be observed normally.

In addition to using foot screws, the balance adjustment of the pendulum body can also be adjusted using the motor inside the pendulum body. The front box is equipped with a motor balance adjustment button, which can be manually adjusted on site or remotely adjusted by logging in to the data collector through the network. The data collector can be powered by DC or AC, and users can access/control the data collector through the RJ45 network port, set network information such as the IP address of the data collector, and complete network access. From the above deployment process, it can be seen that on-site deployment only requires connecting the pendulum body, front box, and data collector with their respective signal lines. The only debugging work is the network setting of the data collector, the unlocking and balancing adjustment of the pendulum body, and the setting of the data acquisition network. The relevant information can be completed by logging in to the data acquisition webpage. Manually rotating the foot screws for balancing adjustment is simple and fast, and the entire process generally does not exceed 10 min, and does not require professional engineering and technical personnel. Ordinary observers can complete it after simple training.

As shown in Figure 7, the pendulum rod has a dedicated positioning groove. The pendulum-locking motor can push the pendulum body against the fixture, and together with the fixture, it clamps the pendulum rod firmly, preventing any movement up, down, left, or right, ensuring the pendulum is locked. After the locking operation, the pendulum can withstand shocks caused by ordinary handling. During transportation, the pendulum is placed in a dedicated transport box with soft foam padding to better withstand bumps and impacts during transit.

## 3. Application of the Rapid Deployment Capability of Integrated Chamber Inclinometers in Geophysical Networks

The rapid deployment capability of the ICT-1 integrated chamber inclinometer not only improves installation efficiency, but also expands the application of high-precision inclinometers in fields such as flow observation, emergency response, and platform detection. In July 2023, the Xinjiang Seismological Bureau was preparing to select a site in the Ili region to install deformation instruments. The Keketai Seismic Cave is located in the outskirts of Xinyuan County, Ili Prefecture, and is equipped with broadband seismometers. Due to its shallow depth of only 20 m and thin coverage, whether it met the requirements of deformation monitoring needed to be concluded based on actual observations. The Xinjiang Seismological Bureau utilized the rapid deployment and stability capabilities of the ICT-1 integrated chamber inclinometer to conduct rapid detection of the Zeketai Mountain cave foundation through actual observations.

### 3.1. Laboratory Calibration

Using a high-precision tilt detection platform, laboratory calibration tests were conducted on the tilt measurement system. The platform was driven to create tilt changes at intervals of 10% of its full scale. Simultaneously, the tilt amount of the platform and the voltage output values of the ICT-1 tilt sensor were recorded. These data were used to calculate parameters such as the instrument’s sensitivity, range, linearity error, and other characteristics.

The laboratory calibration test of the tilt measurement system was conducted using a high-precision tilt detection platform. The structure diagram and physical diagram of the tilt-measuring platform are shown in Figure 8. The measuring platform is installed on the foundation pier. One side is a fixed surface supported by two fixed feet, and the other side is an active surface supported by a piezoelectric ceramic actuator. The piezoelectric ceramic serves as the driving mechanism, capable of generating displacement to lift one side of the platform, thereby inducing a change in tilt. A laser interferometer is employed to record the displacement changes as the platform is lifted, and the tilt change of the platform is calculated from the displacement values. During the test, the pendulum of the ICT-1 tiltmeter is placed on the platform. The platform is driven to produce tilt changes at intervals of approximately 0.2″ (10% of the full-scale value of the tilt test platform). Each time the piezoelectric ceramic lifts the tilt platform, it remains stationary for 2 min before the driving voltage is turned off to allow the platform to return to its original position. After another 2 min of rest, the piezoelectric ceramic increases the displacement to lift the platform again for further testing. The laser interferometer accurately records the lifting height of the platform and converts it into tilt output. The data acquisition system of the tiltmeter records the voltage output of the tiltmeter, as shown in Table 1 and Table 2. This process enables the calculation of parameters such as the sensitivity, range, and linearity error of the tiltmeter. The calibration curve was drawn according to the table data.

Figure 9 presents the calibration test curve for the north–south inclinometer, while Figure 10 provide the calibration test curve for the east–west inclinometer.

### 3.2. Ground Tilt Observation

The Zeketai Seismic Station is located within the north Tianshan latitudinal structural belt, about 20 km from the Kash River fault. The geological formations in the area belong to the Paleozoic, specifically the Devonian and Carboniferous periods. The station and surrounding areas are characterized by metamorphic mudstone and sandstone from the Carboniferous period, with some granite exposures locally. To the south of the station, there is a small-scale east–west trending reverse fault that cuts through the Paleozoic to Late Pleistocene strata. This fault trends east–west, dips steeply to the north, and is classified as a reverse fault, with granite exposed along its length. Given the frequent seismic activity in the Ili region, the Xinjiang Earthquake Bureau intends to install deformation monitoring equipment in this area. The Zeketai Seismic Station, being a seismic observation tunnel, is ideally located in this region. However, the tunnel is relatively shallow and has a cover that is thinner than the required 20 m for observation standards. While excavating a new observation tunnel would be costly and time-consuming, the ICT-1 type inclinometer can be deployed quickly. Thus, a trial observation will be conducted in the Zeketai tunnel to collect actual observation data for station foundation testing. If the data are satisfactory, long-term installation of the inclinometer and other deformation monitoring instruments will be carried out at this site.

Figure 11 shows the original voltage value curve of the ICT-1 integrated chamber inclinometer after being installed for 3 months. From the curve, it can be seen that the ICT-1 integrated chamber inclinometer is also affected by temperature. After the equipment was installed on 15 July 2023, it first experienced significant drift. However, due to its small size and fast heat exchange, the drift slowed down after 10 h of installation, and a clear trend of solid tide can be observed, and the coseismic response can be recorded. On 16 July, the observers remotely zeroed the equipment. By 1 August, the observers determined that the drift had basically ended and formal observation could be carried out.

From the curve in Figure 11, it can be seen that after the official observation on 1 August, the data curve shows clear solid tides, significant changes in tidal magnitude, high signal-to-noise ratio, and good data quality. Table 3 shows the data curve and harmonic analysis results for October 2023. The red dashed box in the figure shows the fluctuation caused by the first data entry into the cave.

From the harmonic analysis results [19,20,21], it can be seen that the observation data for October 2023 have a mean square error of 0.0023 (CH1) and 0.0056 (CH2) for M2 waves, and the data quality meets the technical standard DB/T31.2-2008 for seismic observation instruments entering the network of the China Earthquake Administration [22].

### 3.3. Solid Tide Calculation

Select a relatively smooth time period from the observed curve during the spring tide phase and calculate the theoretical values of the solid tide corresponding to this period. Identify the peak (maximum value point) in the theoretical solid tide value series, denoting it as *d*_0_, and the corresponding time as *t_0_*. Using this point as the center, select 2*n* points (where n is the number of points taken from one side) on either side where the inclination value changes by Δ*d*. Record the corresponding theoretical values and times, denoting them as *d_i_* and *t_i_*, respectively. The change in inclination Δ*d* should be chosen within the range of 0.0001″/2.0 min to 0.0001″/2.5 min [23,24]. 

Based on the selected time period in the theoretical solid tide value series, locate the positions of the peaks in the observed data series within the corresponding time period. According to the time intervals (*T_i_*) of the selected points in the theoretical solid tide series, find the corresponding data points on both sides of the observed data peaks, denoting them as *d*′*ᵢ*. Substitute *d*′*ᵢ* into Formula 10 to calculate the normalization coefficient, denoted as *k*. Multiply the selected points *d*′*ᵢ* from the observed data series by the normalization coefficient *k* to obtain the normalized series, denoted as *d*′*ᵢ*. Substitute *d*″*ᵢ* into Formula 11 to obtain the fitted series, denoted as $\bar{d\prime \prime }ᵢ$. Calculate the difference between the normalized series *d”ᵢ* and the fitted series $\bar{d\prime \prime }ᵢ$ to obtain the difference series Δ*d*″ᵢ. Lastly, select the maximum value between Δ*d*″_−2_ and Δ*d*″_2_; this value represents the instrument’s resolution.(10)$$k=\frac{2\times (n-2)}{\left({d\prime }_{n}-{d\prime }_{2})+({d\prime }_{-n}-{d\prime }_{-2}\right)}\times \frac{0.0001\prime \prime }{0.001\prime \prime }$$(11)$$\bar{d\prime \prime }=\left(\frac{\sum_{i=2}^{n}{d\prime \prime }_{i}}{\left(n-2\right)+1}+\frac{\sum_{i=-2}^{-n}{d\prime \prime }_{i}}{\left(n-2\right)+1}\right)/2$$

According to the testing method in Appendix A of the technical standard DB/T31.2-2008 for earthquake observation instruments entering the network [22], the resolution of tilt measurement is calculated using the solid tide calculation method. Comparing the daily observation data during the spring tide period (16 October 2023) with the theoretical solid tide, the resolution of the measurement system was calculated based on the selected peak data. The resolution of CH1 in the north–south direction was 0.00009″, and the resolution of CH2 in the east–west direction was 0.00017″, as shown in Figure 12 and Figure 13, and Table 4 and Table 5.

From this, it can be concluded that the observation data of the Ketai Mountain Cave meet the requirements of the geophysical network specifications, with a good foundation and the ability to install deformation equipment. At the same time, the data also verify that the ICT-1 integrated chamber inclinometer has the ability to be quickly deployed and has high measurement accuracy, meeting the requirements of high-precision ground tilt observation.

## 4. Discussion

The ICT-1 (Integration Capacitive Tiltmeter) introduced in this paper uses the cross spring as the suspension structure, which successfully solves some key problems of the existing inclinometer. This design not only enhances the strength of the pendulum, but also enables a single pendulum system to simultaneously measure the inclination of multiple components, thereby increasing the measurement efficiency and reducing the equipment volume. To meet the needs of flow observation and emergency measurement, the ICT-1 inclinometer is equipped with a locking mechanism to effectively protect the pendulum. In addition, the use of a short base triangle platform significantly reduces the size of the equipment, while the built-in seal design of the zeroing motor provides protection and shock resistance for general transport. The integrated design also avoids the complexity of on-site assembly and enables rapid deployment. Based on the rapid deployment of ICT-1 inclinometer at Zeketai Seismic Station in Xinjiang, its performance and technical index are analyzed in detail in this paper. The instrument plays an important role in emergency response, mobile observation, platform detection, and anomaly verification of ground tilt, and provides effective support for research and practice in related fields.

## 5. Conclusions

This article introduces the application of the ICT-1 integrated chamber inclinometer in the rapid deployment and base detection of the Xinjiang Geophysical Network. Through on-site deployment and observation, the following conclusions are drawn:

The ICT-1 integrated chamber inclinometer uses a cross-spring vertical pendulum, which enhances the strength of the pendulum body. After installing a self-locking mechanism and a fully enclosed outer cover, it achieves impact resistance during ordinary transportation. The use of a short baseline triangle platform greatly reduces the external dimensions of the equipment. The built-in zero adjustment motor has a compact structure, is easy to carry, does not require on-site assembly, is easy to use, and can be installed without the need for professional technicians. It has the ability to be deployed quickly.The observation resolution of the ICT-1 integrated chamber inclinometer reaches 0.0002”, which is equivalent to the observation resolution of existing chamber pendulum inclinometers and meets the requirements of the seismic observation instrument grid access technical standard DB/T31.2-2008. Like existing chamber tilt inclinometers, the ICT-1 integrated chamber tilt inclinometer is also affected by temperature. During the initial installation of the equipment, there may be significant drift in the observation data. However, due to its small size and fast thermal balance, solid tides can be observed within 24 h of equipment installation and stabilized within a week.The ICT-1 integrated chamber inclinometer has the characteristics of rapid deployment, fast stability, and small size, making it suitable for rapid detection, flow monitoring, emergency response, and other fields on platforms. It is of great significance for the observation of geophysical field deformation.

## Figures and Tables

**Figure 1 sensors-25-01559-f001:**
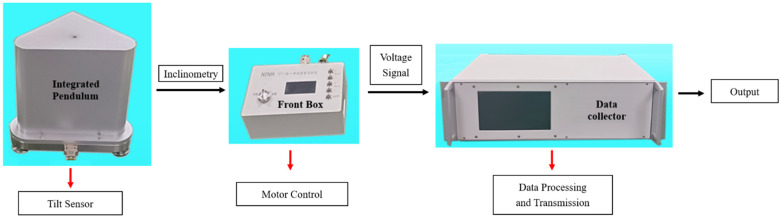
Schematic diagram of high-precision integrated chamber inclinometer.

**Figure 2 sensors-25-01559-f002:**
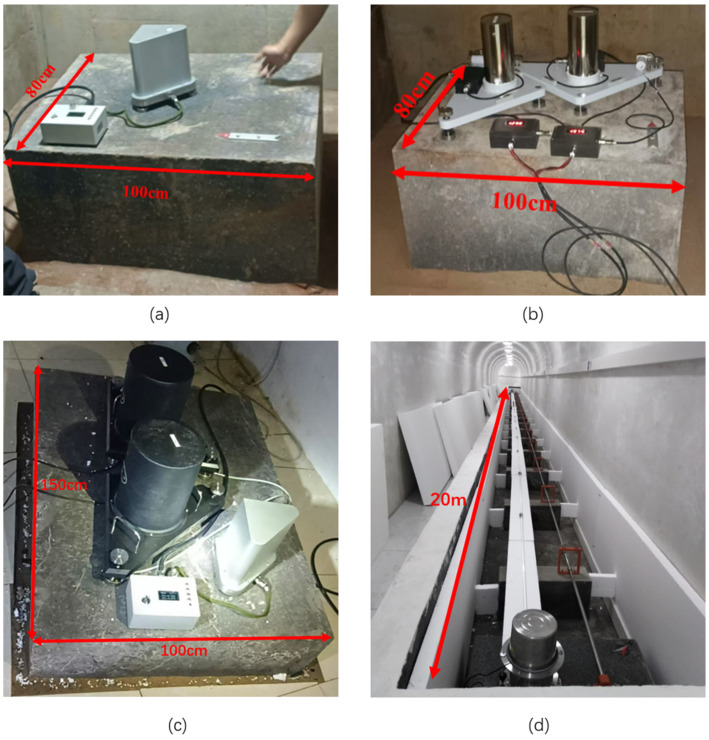
Installation diagram of ICT-1 integrated inclinometer, VP Single-component vertical pendulum inclinometer, SQ-70D quartz horizontal pendulum inclinometer, and DSQ-type water pipe inclinometer ((**a**) shows ICT-1 integrated inclinometer; (**b**) shows VP single-component vertical pendulum inclinometer; (**c**) shows a comparison of ICT-1 integrated inclinometer and SQ-70D quartz horizontal pendulum inclinometer, in which the two black instruments are the quartz horizontal pendulum; (**d**) shows DSQ-type water pipe inclinometer).

**Figure 3 sensors-25-01559-f003:**
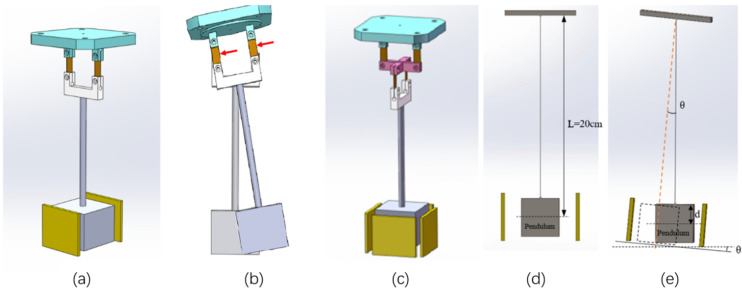
Schematic diagram of inclined pendulum system. (**a**) is the dual-leaf spring suspension structure. (**b**) is when the dual-leaf spring suspension structure receives an impact in a non-moving direction, the spring blade will be damaged, and the red arrow is the direction of the impact force. (**c**) is the cross-spring pendulum structure. (**d**,**e**) represent a unidirectional pendulum and its motion process.).

**Figure 4 sensors-25-01559-f004:**
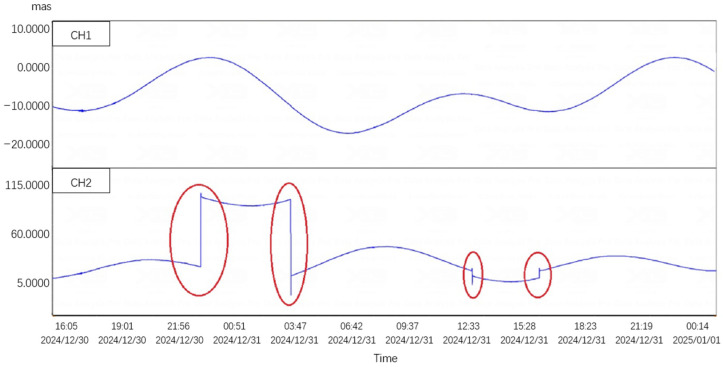
The phenomenon of skipping frames that occurred with the CBT-1 inclinometer using a unidirectional spring blade at the Xinyuan seismic station in Xinjiang on 30 December 2024. The red circle in the figure shows the phenomenon of skipping frames in the experiment.

**Figure 5 sensors-25-01559-f005:**
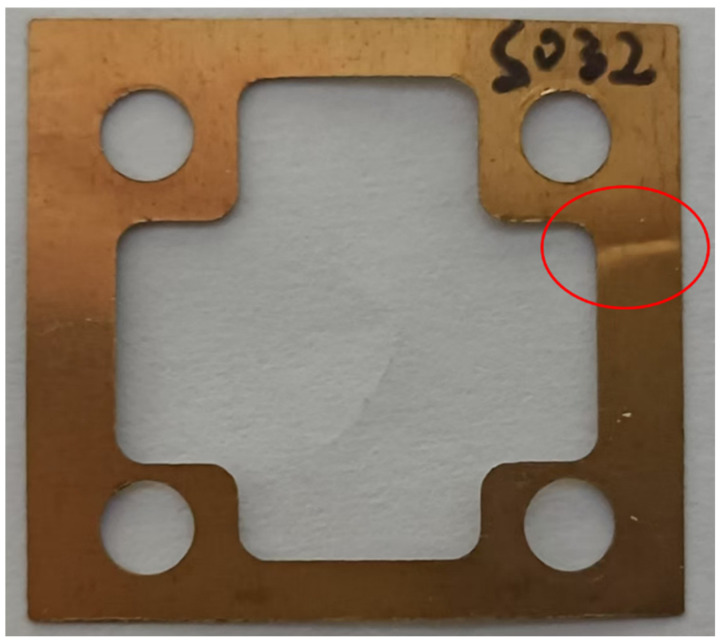
Physical image of the CBT-1 inclinometer’s spring blade after being impacted (the red circle indicates the location of the crease).

**Figure 6 sensors-25-01559-f006:**
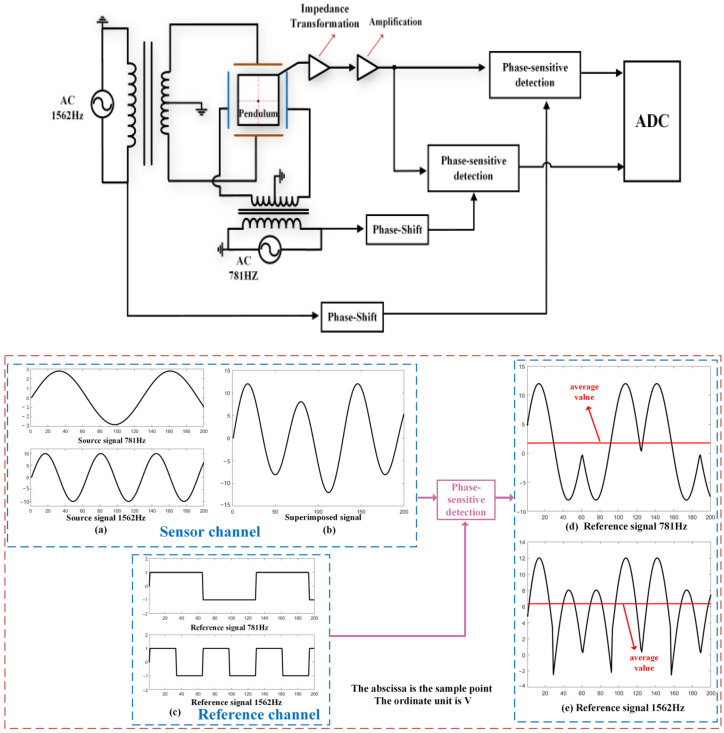
Electrical schematic diagram of integrated chamber inclinometer. Subfigure (**a**) shows the unbalanced signals of two groups of differential Bridges. Subfigure (**b**) shows the superposition of two sets of differential bridge unbalanced signals on the pendulum. Subfigure (**c**) shows that the excitation signals of the two groups of Bridges are transformed into square waves after shaping, which is the waveform of the reference signal. Subfigures (**d**,**e**) show the waveforms of the two groups of differential Bridges after being detected by the phase-sensitive detection circuit using different reference signals.

**Figure 7 sensors-25-01559-f007:**
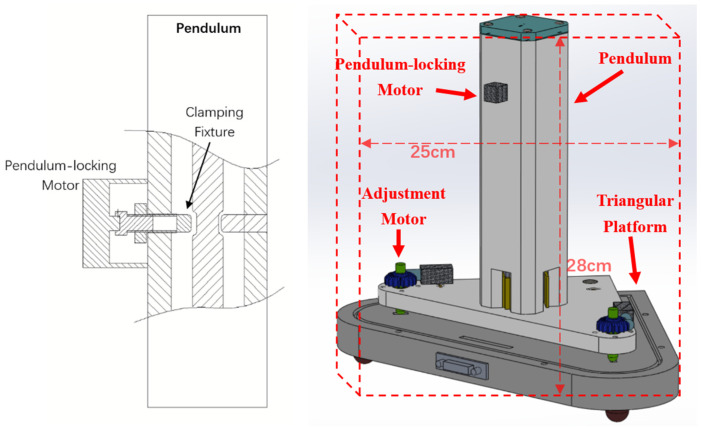
Structural diagram of the pendulum-locking motor (**left** figure) and schematic diagram of the ICT-1 tiltmeter triangular platform (**right** figure).

**Figure 8 sensors-25-01559-f008:**
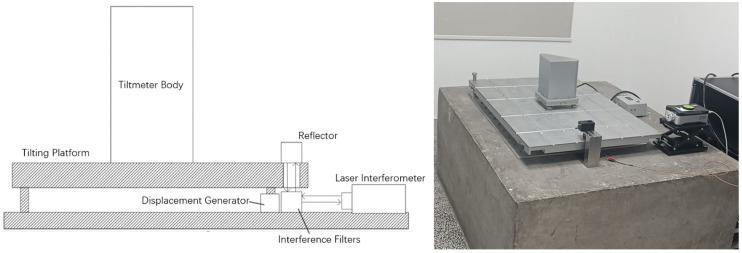
Diagram of the tilting measurement platform structure and physical appearance.

**Figure 9 sensors-25-01559-f009:**
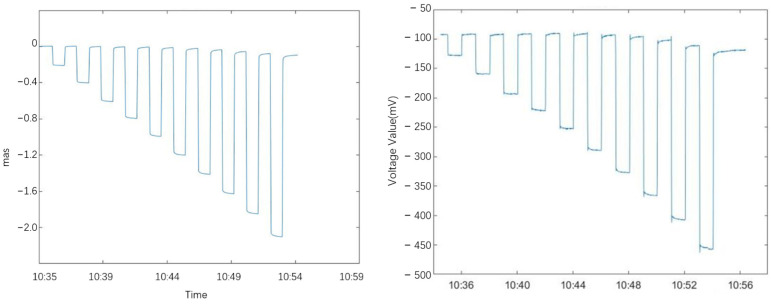
North–South Direction Tilt Platform Changes and Inclinometer Voltage Output Variation Chart.

**Figure 10 sensors-25-01559-f010:**
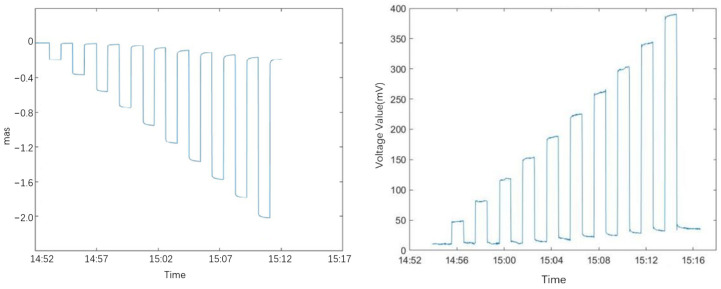
East–West Direction Tilt Platform Changes and Inclinometer Voltage Output Variation Chart.

**Figure 11 sensors-25-01559-f011:**
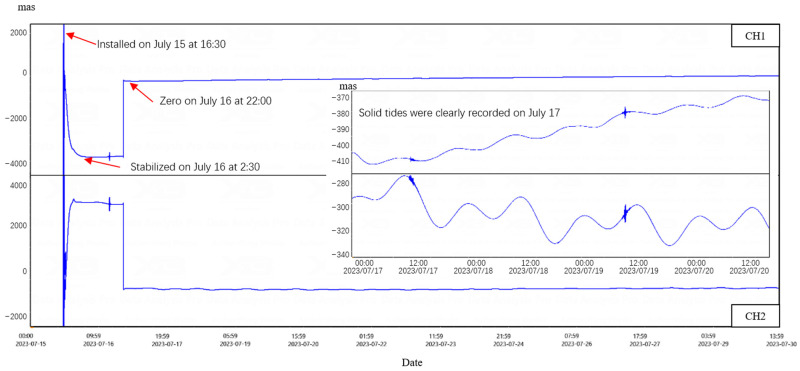
Observation curve of inclinometer at Zeketai Seismic Station in Xinjiang. The protrusion of CH1 on August 8 was used to collect data from the cave.

**Figure 12 sensors-25-01559-f012:**
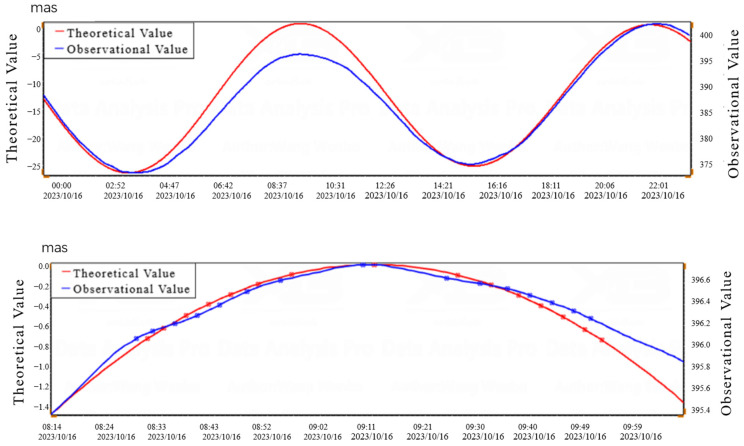
North–South Direction Theoretical and Observational Value Curve on 16 October 2023.

**Figure 13 sensors-25-01559-f013:**
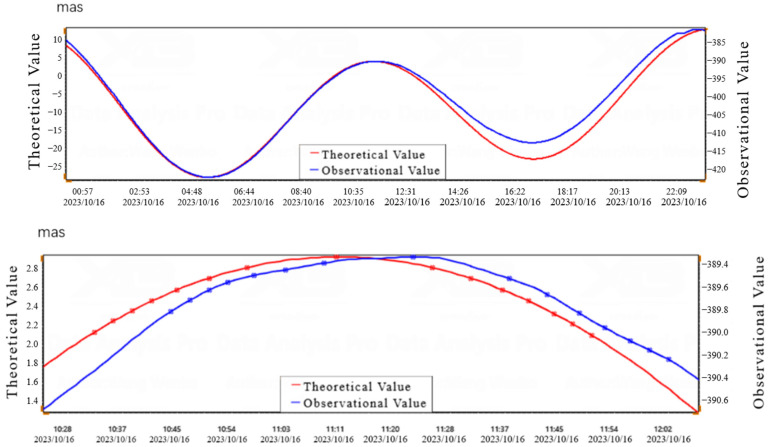
East–West Direction Theoretical and Observational Value Curve on 16 October 2023.

**Table 1 sensors-25-01559-t001:** North-South Inclinometer Calibration Data.

No.	Tilt Amount $x_{i}$ (″)	Sensor Output $y_{i}$ (mV)	Fitted Value $Y_{i}$ (mV)	Linear Deviation $\Delta y_{i}$
1	0.21859	35.555	33.075	2.4799
2	0.4169	67.731	65.97	1.761
3	0.62531	100.89	100.54	0.35175
4	0.81379	130.11	131.81	1.6982
5	1.0145	161.47	165.1	3.6355
6	1.2229	197.48	199.66	2.1857
7	1.4317	233.2	234.3	1.0987
8	1.6388	269.32	268.67	0.6571
9	1.8474	304.44	303.26	1.1824
10	2.0854	344.93	342.74	2.1859
	$x=\frac{\Delta H}{L}\rho$	$\Delta y_{FS}$ = 344.93	d = 0.34%	

**Table 2 sensors-25-01559-t002:** East–West Inclinometer Calibration Data.

No.	Tilt Amount $x_{i}$ (″)	Sensor Output $y_{i}$ (mV)	Fitted Value $Y_{i}$ (mV)	Linear Deviation $\Delta y_{i}$
1	0.2202	37.2903	36.6965	0.5938
2	0.4116	68.638	69.0464	0.4084
3	0.6209	107.6002	104.4043	3.196
4	0.826	141.083	139.063	2.02
5	1.0348	173.3161	174.3381	1.022
6	1.2417	205.6679	209.2865	3.6186
7	1.4433	238.1662	243.3571	5.1909
8	1.6491	277.2836	278.1268	0.8433
9	1.852	313.1923	312.4131	0.7792
10	2.086	356.4319	351.9378	4.4942
	$x=\frac{\Delta H}{L}\rho$	$\Delta y_{FS}$ = 356.4319	d = 0.47%	

**Table 3 sensors-25-01559-t003:** Venidikov reconciliation analysis of tilt observation data from Zeketai Seismic Station in October 2023.

Strain Channel	Tidal Factor	Tidal Factor Mean Squared Error	Tidal Phase Lag	Tidal Phase Lag Mean Squared Error
CH1 (NS)	0.3105	0.0023	−1.5667	−0.2473
CH2 (EW)	0.3918	0.0056	−4.8036	0.2587

**Table 4 sensors-25-01559-t004:** North–South Direction Resolution calculation table for the solid tidal strain gauge.

N	Date	Theory Value	Time Interval	Obs Value	Norm Value	Fit Value	Difference
	16 October 2023	*dᵢ*	Ƭᵢ	*d*′*ᵢ*	*d*″*ᵢ*	$\bar{d\prime \prime }ᵢ$	Δ*d*″*ᵢ*
	Time	0.001″	min	0.001″	0.001″	0.001″	0.001″
−7	08:31	−0.6423	41	396.0720	508.4365	508.5004	0.0639
−6	08:34	−0.5322	38	396.1450	508.5302	508.6201	0.0899
−5	08:38	−0.3981	34	396.2140	508.6187	508.7398	0.1210
−4	08:42	−0.2788	30	396.2920	508.7189	508.8595	0.1406
−3	08:46	−0.1742	26	396.3910	508.8460	508.9792	0.1332
−2	08:51	−0.0645	21	396.5200	509.0116	509.0989	0.0873
−1	08:57	0.0362	15	396.6240	509.1451	509.2186	0.0735
0	09:12	0.1391	0	396.7730	509.3363	509.3383	0.0020
1	09:27	0.0291	15	396.6470	509.1746	509.2186	0.0440
2	09:33	−0.0740	21	396.5930	509.1053	509.0989	0.0064
3	09:38	−0.1853	26	396.5440	509.0424	508.9792	0.0632
4	09:42	−0.2908	30	396.4830	508.9641	508.8595	0.1046
5	09:46	−0.4109	34	396.4120	508.8729	508.7398	0.1331
6	09:50	−0.5453	38	396.3310	508.7689	508.6201	0.1489
7	09:53	−0.6553	41	396.2620	508.6804	508.5004	0.1800
K		1.2837		Δ*d*″max		0.0883	

**Table 5 sensors-25-01559-t005:** East–West Direction Resolution calculation table for the solid tidal strain gauge.

N	Date	Theory Value	Time Interval	Obs Value	Norm Value	Fit Value	Difference
	16 October 2023	*dᵢ*	Ƭᵢ	*d*′*ᵢ*	*d*″*ᵢ*	$\bar{d\prime \prime }ᵢ$	Δ*d*″*ᵢ*
	Time	0.001″	min	0.001″	0.001″	0.001″	0.001″
−7	10:33	2.1379	38	−389.8900	−399.4775	−399.7685	0.2910
−6	10:36	2.2571	35	−389.7870	−399.3719	−399.6451	0.2731
−5	10:39	2.3668	32	−389.6970	−399.2797	−399.5217	0.2419
−4	10:42	2.4670	29	−389.6250	−399.2059	−399.3983	0.1923
−3	10:46	2.5856	25	−389.5580	−399.1373	−399.2749	0.1376
−2	10:51	2.7097	20	−389.5100	−399.0881	−399.1515	0.0633
−1	10:57	2.8229	14	−389.4490	−399.0256	−399.0281	0.0024
0	11:11	2.9354	0	−389.3910	−398.9662	−398.9047	0.0615
1	11:26	2.8212	15	−389.5900	−399.1701	−399.0281	0.1420
2	11:32	2.7084	21	−389.7340	−399.3176	−399.1515	0.1662
3	11:37	2.5856	26	−389.9030	−399.4908	−399.2749	0.2159
4	11:41	2.4687	30	−390.0410	−399.6322	−399.3983	0.2339
5	11:45	2.3355	34	−390.1560	−399.7500	−399.5217	0.2283
6	11:48	2.2249	37	−390.2440	−399.8402	−399.6451	0.1951
7	11:51	2.1054	40	−390.3300	−399.9283	−399.7685	0.1598
K		1.0246		Δ*d*”_max_		0.1657	

## Data Availability

Data are contained within the article.

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
