# Peer review of "Application Research on High-Precision Tiltmeter with Rapid Deployment Capability"

_sensors, 2025, doi:10.3390/s25051559_

Round 1

Reviewer 1 Report

Comments and Suggestions for Authors

Author Response

Comment 1: Figure 1: Include a photograph of the fully assembled instrument, preferably installed in a field setting, to provide a clearer understanding of its design and application.

Response 1: Thank you for your suggestions. I have added a comparison diagram of the instrument installation site in the second section (see Figure 2).

Comment 2: Figure 2: Add dimensions or a scale to improve the interpretability of the schematic.

Response 2: Thank you for your suggestion. I have added dimension labels to Figure 2 (now Figure 3).

Comment 3: Dual-Leaf Spring Suspension Structure: The manuscript states that the dual-leaf spring suspension structure is superior to the dual-suspension leaf springs arranged in orthogonal

superposition. However, this claim must be substantiated with experimental data. Assertions without ground-truth support are insuƯ icient.

Response 3: Thank you for your suggestion. I have included relevant content, experimental curves, and physical images in Section 2.1 of the text. The dual-spring suspension structure has measuring sensitivity comparable to that of the orthogonal stacked dual suspension plates, but with enhanced impact resistance, preventing skipping issues. It is also convenient for installation and transport.

Comment 4: Calibration and Validation: It is essential to include laboratory-based calibration or

validation of the instrument prior to field deployment. Characteristics such as measuring range, resolution, and accuracy should be thoroughly analysed and reported.

Response 4: Thank you for your suggestion. I have supplemented the calibration experiment in Section 3.1 of the third part of the text, hoping to address your question.

Comment 5: Temperature Interference (Figure 4): The temperature-induced variation in sensor output, showing a shift of up to 3 Volts, is questionable. The authors should clarify the temperature

ranges that led to this variation. Additionally, after zeroing the sensor, one would expect output variations due to temperature gradients between day and night. This aspect should be analysed further.

Response 5: Thank you for your opinion. A high-precision inclinometer requires the ability to observe clear solid tidal effects, which typically cause ground tilt changes of about 0.01 to 0.04″. Therefore, the network standards require that the resolution of the inclinometer measurements in the chamber reach 0.0002″. At such observation levels, temperature significantly affects measurements. Thus, the standards specify that the cover thickness at the top of the chamber should not be less than 20m, with daily temperature variations not exceeding 0.03℃ and annual temperature changes not exceeding 0.5℃. During the initial installation phase, the inclinometer enters the chamber from outside, causing a significant temperature shift before thermal equilibrium is reached. I hope this addresses your question.

Comment 6: Figure 4 - Readability and Units: The inset image labels are too small to read. The sensor output during field experiments should be expressed in tilt units (e.g., radians or degrees) rather than electrical potential. As stated before, a laboratory calibration is required to correlate the sensor’s output to actual tilt magnitudes.

Response 6: Thank you for your suggestion. I have supplemented the calibration experiment in Section 3.1 of the text, and provided tabular data and curves to relate the sensor output to the actual tilt amplitude. I have also redrawn Figure 4 (now Figure 9), improving its clarity and readability, hoping to resolve your question.

Comment 7: Figure 5: Specify the units on the vertical axis. Address why the signals are not in phase and provide an explanation.

Response 7: Thank you very much for identifying the issues. Following your suggestion, I have added the units for the vertical axis in Figures 5 (now Figures 10 and 11). Because the equipment used in the Zekitai observation experiment was not connected to the network, the data acquisition system could not use network or GPS time calibration, resulting in the data acquisition clock being nearly 25 minutes behind the actual time. This is the main reason for the phase difference. After time calibration, the actual observation values closely match the theoretical solid tide phase. However, the calculation of the theoretical solid tide assumes that the Earth is a uniform elastic sphere, without considering the effects of faults, terrain, elevation differences, lithology, etc. Therefore, even after time calibration, there is still some deviation between the theoretical values and the observed values.

Comment 8: Figure Captions: Rephrase all figure captions to provide more context and detail. Captions should allow readers to fully understand the figures without needing to refer back to the main text.

Response 8: Thank you for identifying the issues. I have revised all unclear descriptions in the figures.

Reviewer 2 Report

Comments and Suggestions for Authors

This is a meaningful research topic, However, there are still several problems in the manuscript that need to be revised and improved:

1The ordinate of figure 5 has no unit.

2The observation resolution of the ICT-1 integrated chamber inclinometer reaches 0.0002 ",However, it lacks of the static characteristics test process and results, please supplement and improve.

Author Response

Comment 1: The ordinate of figure 5 has no unit.

Response 1: Thank you very much for identifying the issues. Following your suggestion, I have added the units for the vertical axis in Figures 5 (now Figures 10 and 11).

Comment 2: The observation resolution of the ICT-1 integrated chamber inclinometer reaches 0.0002",However, it lacks of the static characteristics test process and results, please supplement and improve.

Response 2: Thank you for your suggestions. I have supplemented the calibration experiment in Section 3.1 of the text, and provided tabular data and curves to relate the sensor output to the actual tilt amplitude. In geological measurements, high-precision inclinometers must achieve a resolution of 0.0002". Currently, the available inclination calibration platforms cannot provide such high tilt values. Therefore, geological observations employ theoretical solid tide calculation methods to assess the resolution of the equipment. I have added the process of solid tide calculation, along with the measured curves and data tables, in Section 3.3 of the text (Figures 10 and 11, Tables 4 and 5). Hoping to address your question.

Reviewer 3 Report

Comments and Suggestions for Authors

This paper presents a high-precision vertical pendulum tilt meter developed for rapid deployment, aiming to enhance the efficiency and practicality of geophysical site tilt observations. The device employs a vertical pendulum suspended by a cross spring, utilizing a differential capacitance bridge to measure ground tilt in two orthogonal directions. Its effectiveness is validated through application at the Zeketai seismic station in Xinjiang, highlighting its significance for emergency response, mobile observation, and other ground tilt monitoring applications. However, there are several aspects of the current structure and presentation of the manuscript that require clarification for publication. Below are specific comments:

1.    Line 33: Please include a more detailed comparative analysis between the ICT-1 and older tiltmeters.

2.    Line 43: Please provide additional information regarding the ICT-1's innovations, specifically the cross-spring suspension system, and its impact on data collection in the field.

3.    Line 187: The description of the deployment process highlights ease of use and speed, which are critical features of the ICT-1 inclinometer. However, the deployment steps could benefit from greater emphasis on the flexibility and adaptability of the system under various field conditions.

4.    Line 198: The process mentions adjusting the foot screws to achieve equilibrium, but please clarify how sensitive this adjustment is.

5.    Line 203: Please highlight the specific advantages of remote adjustment, such as the capability to modify settings from a central command post during emergencies, or how remote access enhances real-time monitoring in critical situations.

6.    Line 210: Please elaborate on the level of training required for ordinary observers. What does the training involve? Furthermore, how automated is the process?

7.    Line 220: Please explain why the Keketai Seismic Cave was selected for testing the ICT-1. Was it chosen due to specific geological conditions, the presence of active seismic fault lines, or as a representative site for deformation monitoring?

8.    Line 233: Please provide more quantitative data or examples of the observed drift, including how long it took to stabilize. Additionally, explain how this temperature sensitivity is typical for such instruments and how the rapid heat exchange of the ICT-1 helps mitigate long-term drift. Is this drift an issue that requires regular calibration, or is it a one-time event during initial setup?

9.    Line 244: Explain why a high signal-to-noise ratio is important in this context. Does it reflect the inclinometer’s ability to detect subtle changes in tilt with greater precision? How does this performance compare to other systems in the field, and what implications does it have for the broader geophysical network?

10. Line 257: Explain how these resolution values align with or exceed the precision required for typical deformation monitoring in geophysical networks. Are these values adequate for monitoring tectonic movement, fault lines, or seismic precursors? How do they compare with the precision offered by other types of tiltmeters or inclinometer systems?

Author Response

Comment 1: Line 33: Please include a more detailed comparative analysis between the ICT-1 and older tiltmeters.

Response 1: Thank you for your suggestion. I have added a comparison diagram of the instrument installation site in the second section (see Figure 2). The dual-spring suspension structure has measuring sensitivity comparable to that of the orthogonal stacked dual suspension plates, but with enhanced impact resistance, preventing skipping issues. I have included relevant content, experimental curves, and physical images in Section 2.1 of the text. It is also convenient for installation and transport.

Comment 2: Line 43: Please provide additional information regarding the ICT-1's innovations, specifically the cross-spring suspension system, and its impact on data collection in the field.

Response 2: Thank you for your suggestion. The ICT-1 has a measurement sensitivity comparable to other inclinometers, such as the VP single-component vertical pendulum inclinometer. However, it features enhanced shock resistance, eliminating issues like skipping or jumping readings. Additionally, it is convenient to install and carry. The text also provides installation site images for the two types of inclinometers(see Figure 2). A comparison shows that the ICT-1 is more compact and portable.

Comment 3: Line 187: The description of the deployment process highlights ease of use and speed, which are critical features of the ICT-1 inclinometer. However, the deployment steps could benefit from greater emphasis on the flexibility and adaptability of the system under various field conditions.

Response 3: Thank you for your question. The ICT-1 type inclinometer is designed for the same application environment but enhances rapid deployment capabilities. Its operating environment and requirements for observation conditions are the same as those of other tunnel instruments.

Comment 4: Line 198: The process mentions adjusting the foot screws to achieve equilibrium, but please clarify how sensitive this adjustment is.

Response 4: Thank you for your question. The platform uses a stepper motor with a gear ratio of 1:100 as a fine-tuning mechanism for zeroing. One full rotation of the motor corresponds to 20,000 steps, allowing the platform to tilt approximately 0.06°. With a 100-step adjustment, each step can tilt the platform by 1.074″, achieving the required calibration accuracy. However, the adjustment speed of the stepper motor is relatively slow, taking about 3 minutes for one full rotation.

To facilitate quick zeroing during initial installation and enable rapid deployment, manual rotation screws are set at the bottom of the platform for coarse adjustment. Each full turn of these screws can tilt the platform by about 0.6°. In practice, the bottom screws are first manually rotated to quickly bring the pendulum to a near-balanced position, after which the stepper motor is used for precise adjustments.

Comment 5: Line 203: Please highlight the specific advantages of remote adjustment, such as the capability to modify settings from a central command post during emergencies, or how remote access enhances real-time monitoring in critical situations.

Response 5: Thank you for your question. The data collector is equipped with a network protocol, allowing for remote access via web login to modify operating parameters such as filtering constants and sampling rates, or to download data. It can also remotely drive the motor for zeroing operations. In emergency monitoring situations, it can conduct encrypted sampling and provide real-time data monitoring.

Comment 6: Line 210: Please elaborate on the level of training required for ordinary observers. What does the training involve? Furthermore, how automated is the process?

Response 6: Thank you for your question. The training for ordinary observers includes:

  1. Correct unboxing and carrying of the equipment.
  2. Placement of the pendulum on the observation platform, connection of the signal cables, power cables, and network cables to the front box and data collector, as well as turning the equipment on/off and unlocking the pendulum.
  3. Manual leveling of the base screws of the inclined platform and fine leveling using the stepping motor.
  4. Setting the network and operating parameters of the equipment and zeroing the travel.

The training method is hands-on practice according to the user manual. Mastering these four points will enable proficient installation and use of the equipment, typically achievable within 30 minutes of training and practice.

Comment 7: Line 220: Please explain why the Keketai Seismic Cave was selected for testing the ICT-1. Was it chosen due to specific geological conditions, the presence of active seismic fault lines, or as a representative site for deformation monitoring?

Response 7: Thank you for your question. The Zeketai Seismic Station is located within the north Tianshan latitudinal structural belt, about 20 kilometers from the Kash River fault. The geological formations in the area belong to the Paleozoic, specifically the Devonian and Carboniferous periods. The station and surrounding areas are characterized by metamorphic mudstone and sandstone from the Carboniferous period, with some granite exposures locally.To the south of the station, there is a small-scale east-west trending reverse fault that cuts through the Paleozoic to Late Pleistocene strata. This fault trends east-west, dips steeply to the north, and is classified as a reverse fault, with granite exposed along its length. Given the frequent seismic activity in the Ili region, the Xinjiang Earthquake Bureau intends to install deformation monitoring equipment in this area. The Zeketai Seismic Station, being a seismic observation tunnel, is ideally located in this region. However, the tunnel is relatively shallow and has a cover that is thinner than the required 20 meters for observation standards. While excavating a new observation tunnel would be costly and time-consuming, the ICT-1 type inclinometer can be deployed quickly. Thus, a trial observation will be conducted in the Zeketai tunnel to collect actual observation data for station foundation testing. If the data is satisfactory, long-term installation of the inclinometer and other deformation monitoring instruments will be carried out at this site. I have added the content to section 3.2 of the article on solid tide observations, and I hope this can answer your question.

Comment 8: Line 233: Please provide more quantitative data or examples of the observed drift, including how long it took to stabilize. Additionally, explain how this temperature sensitivity is typical for such instruments and how the rapid heat exchange of the ICT-1 helps mitigate long-term drift. Is this drift an issue that requires regular calibration, or is it a one-time event during initial setup?

Response 8: Thank you for your question. High-precision chamber inclinometers need to observe clear solid tidal effects, which typically cause a daily variation in ground tilt ranging from 0.01 to 0.04″. Therefore, the observational standards set by the geophysical network require that the resolution of chamber tilt measurements reach 0.0002″. At this level of observation, temperature has a significant impact on measurements; thus, strict environmental conditions are imposed. The standards specify that the thickness of the covering at the top of the chamber must be no less than 20 meters, with daily temperature variations within the chamber not exceeding 0.03℃ and annual temperature changes not exceeding 0.5℃.

During the initial installation of the equipment, the inclinometer enters the chamber from the outside, leading to a significant temperature difference between the device and the environment. Before reaching thermal equilibrium, this can result in considerable temperature drift. This characteristic is common to all high-precision chamber deformation devices and is a one-time event during the initial installation process. The ICT-1 chamber inclinometer, being relatively compact, allows for faster thermal exchange. Additionally, its greatest advantage is its ease of installation and quick deployment, which means that the impact of temperature disturbances within the chamber is minimized. Consequently, the equipment stabilizes more rapidly; however, this does not alleviate the long-term drift caused by environmental temperature changes.

In the early stages of installation, there will always be considerable temperature drift, which is influenced by the temperature difference between the inside and outside of the chamber, the thickness of the chamber covering, the lithology of the chamber, and humidity levels. Even the same device may perform differently when installed in various chambers or seasons. The geophysical network standards require that instruments installed in chambers undergo a stabilization period of two months, during which temperature changes inside the chamber must be monitored as a reference parameter. Currently, it is challenging to quantify or calibrate temperature drift, and this paper does not cover related research on this topic.

Comment 9: Line 244: Explain why a high signal-to-noise ratio is important in this context. Does it reflect the inclinometer’s ability to detect subtle changes in tilt with greater precision? How does this performance compare to other systems in the field, and what implications does it have for the broader geophysical network?

Response 9: Thank you for your question. The reference to a high signal-to-noise ratio in the text is relative to solid tides. The daily variation in ground tilt caused by solid tides ranges from 0.01 to 0.04″, while the geophysical network requires high-precision chamber inclinometers to achieve a measurement resolution of 0.0002″. The higher the resolution of the equipment, the smaller the noise compared to the solid tide wave signal (resulting in a high signal-to-noise ratio), allowing for clearer recordings of solid tidal waves. Some observation stations experience higher noise levels because their equipment resolution does not reach 0.0002″ or due to significant interference sources nearby, leading to a lower signal-to-noise ratio and poorer quality recordings of solid tides. This serves as an empirical and intuitive evaluation of the sensitivity of the chamber deformation equipment.

Do you think it is necessary to remove the mention of the signal-to-noise ratio, or should we add a note stating that the noise is much smaller than the solid tide waves, allowing for clear recordings of solid tides?

Comment 10: Line 257: Explain how these resolution values align with or exceed the precision required for typical deformation monitoring in geophysical networks. Are these values adequate for monitoring tectonic movement, fault lines, or seismic precursors? How do they compare with the precision offered by other types of tiltmeters or inclinometer systems?

Response 10: Thank you for your question. The China Earthquake Networks Center, based on years of monitoring research from the geophysical network, has established relevant industry standards, stating that "the resolution of chamber tilt observations should be ≤0.0002″." This resolution is sufficient to observe clear solid tidal effects, tectonic movements, and fault activities. Various chamber tilt instruments, including the ICT-1 type, VP type, and VS type, are all set to a resolution of 0.0002″.

The ICT-1 type inclinometer does not differ significantly from other types in terms of measurement accuracy and resolution. Its innovation and features lie in the use of a cross spring in the pendulum, which reduces size while enhancing the flexibility of oscillation in all directions and improving impact resistance. This makes it more portable, easier to operate, and allows for rapid deployment and emergency monitoring.

Round 2

Reviewer 1 Report

Comments and Suggestions for Authors

Comments on the Quality of English Language

The language is generally clear and straightforward, but some sections feel a bit rough. A thorough revision of the manuscript would be appreciated.

Author Response

Comment 1: The manuscript provides a very limited review of existing literature. Only a handful of tilt instruments are mentioned, and some date back more than a century. The authors must perform a comprehensive review of recent advancements in tilt instrumentation. This should include analysing the novelty, advantages, and disadvantages of existing instruments, to better contextualize their contribution and demonstrate the gap their work aims to address.

Response 1: Thanks for your suggestion, I have added a variety of inclinometers currently in use in the introduction according to your suggestion, and introduced the principle, background, advantages and disadvantages of each type of inclinometer, and added the installation site diagram (Figure 2), hoping to answer your questions.

Comment 2: The research questions and objectives are not clearly articulated. Furthermore, the final paragraph (Lines 68–82) is a summary of the study and would be better suited for the Conclusion section.

Response 2: Thanks for your suggestion, I have changed the last paragraph (Line 68-82) to the "Discussion" part, and at the same time explained the shortcomings of the current use of the inclinometer in the introduction part, aiming at these shortcomings, led to the ICT-1 component vertical pendulum inclinometer designed in this paper.

Comment 3: The discussion fails to adequately highlight the novelty, achievements, limitations, and potential future directions of the study. Moreover, this section should include a direct comparison of the presented instrument with existing alternatives, clearly demonstrating its uniqueness and advantages.

Response 3: Thanks for your suggestions, I have rewritten the content of the " Discussion " part and added the "Conclusions" part.

Reviewer 3 Report

Comments and Suggestions for Authors

This paper presents a high-precision vertical pendulum tilt meter developed for rapid deployment. The device aims to improve the efficiency and ease of geophysical site tilt measurements. It features a vertical pendulum suspended by a cross spring, using a differential capacitance bridge to measure ground tilt in two perpendicular directions. This subsequent review still lacks the necessary clarity required for publication. Please find below comments and suggestions for enhancing the quality of the manuscript:

  1. Line 89: Please consider expanding on the technical details of the locking motor mechanism to enhance the understanding of how the inclinometer is physically secured during transportation. Additionally, please provide more context on the operation of differential capacitance sensing in this scenario and the rationale for selecting it over other sensor technologies.
  2. Line 118: Please elaborate on the reasons why unidirectional pendulums are not suited for portable or emergency use. Additionally, provides more detail on the performance of the unidirectional design under rapid or varying conditions.
  3. Line 267: The calibration methodology requires a more detailed explanation, including how the tilt angle and corresponding voltage signals were recorded and processed.
  4. Line 321: Please clarify the rationale for utilizing this specific range of tilt change (0.0001″/2.0 min to 0.0001″/2.5 min) and whether it has been optimized for the measurement system.
  5. Line 369: Separate the Discussion and Conclusion into two distinct sections.

Author Response

Comment 1: Line 89: Please consider expanding on the technical details of the locking motor mechanism to enhance the understanding of how the inclinometer is physically secured during transportation. Additionally, please provide more context on the operation of differential capacitance sensing in this scenario and the rationale for selecting it over other sensor technologies.

Response 1: Thanks for your suggestion, I have added the background of the differential capacitance sensor in section 2.2 of the article and the reasons for choosing it over other sensors; The technical details of the locking motor mechanism and the structure diagram (Figure 7) have been added in section 2.3 of the article, hoping to answer your questions.

Comment 2: Line 118: Please elaborate on the reasons why unidirectional pendulums are not suited for portable or emergency use. Additionally, provides more detail on the performance of the unidirectional design under rapid or varying conditions.

Response 2: Thanks for your suggestion, I have added more technical details and selection reasons of unidirectional pendulum in section 2.1 of the article, and updated Figure 3 to facilitate understanding, hoping to answer your questions

Comment 3: Line 267: The calibration methodology requires a more detailed explanation, including how the tilt angle and corresponding voltage signals were recorded and processed.

Response 3: Thank you for your suggestions. I have added the relevant details of the calibration process in Section 3.1 of the article, including the structure diagram and physical diagram of the measurement platform, the measurement method and process, etc. I hope I can answer your questions.

Comment 4: Line 321: Please clarify the rationale for utilizing this specific range of tilt change (0.0001″/2.0 min to 0.0001″/2.5 min) and whether it has been optimized for the measurement system.

Response 4: Thanks for the advice, I added to your response Technical requirements of instruments in network for earthquake monitoring. The instrument for crustal deformation observation.  Part 1: Tiltmeter.  The standards in this paper are strictly implemented in accordance with the attached standards. I hope the attachment can answer your question.

Comment 5: Line 369: Separate the Discussion and Conclusion into two distinct sections.

Response 5:  Thanks for your suggestions, I have rewritten the content of the " Discussion " part and added the "Conclusions" part.